# Resting State EEG Correlates of Suicide Ideation and Suicide Attempt

**DOI:** 10.3390/jpm13060884

**Published:** 2023-05-24

**Authors:** Francesco Amico, Richard E. Frye, Scott Shannon, Steve Rondeau

**Affiliations:** 1Neotherapy, Second Level, 2225 N Commerce Pkwy Suite #6, Weston, FL 33326, USA; dr.amico@txlifemed.org; 2Texas Center for Lifestyle Medicine, 333 West Loop N. Ste 250, Houston, TX 77024, USA; 3Autism Discovery and Treatment Foundation, Phoenix, AZ 85050, USA; 4Department of Psychiatry, University of Colorado, Aurora, CO 80045, USA; 5Wholeness Center, 2620 East Prospect Road, #190, Fort Collins, CO 80525, USA; 6Axon EEG Solutions, Fort Collins, CO 80528, USA

**Keywords:** suicide risk assessment, resting state, electroencephalogram, EEG

## Abstract

Suicide is a global phenomenon that impacts individuals, families, and communities from all income groups and all regions worldwide. While it can be prevented if personalized interventions are implemented, more objective and reliable diagnostic methods are needed to complement interview-based risk assessments. In this context, electroencephalography (EEG) might play a key role. We systematically reviewed EEG resting state studies of adults with suicide ideation (SI) or with a history of suicide attempts (SAs). After searching for relevant studies using the PubMed and Web of Science databases, we applied the PRISMA method to exclude duplicates and studies that did not match our inclusion criteria. The selection process yielded seven studies, which suggest that imbalances in frontal and left temporal brain regions might reflect abnormal activation and correlate with psychological distress. Furthermore, asymmetrical activation in frontal and posterior cortical regions was detected in high-risk depressed persons, although the pattern in the frontal region was inverted in non-depressed persons. The literature reviewed suggests that SI and SA may be driven by separate neural circuits and that high-risk persons can be found within non-depressed populations. More research is needed to develop intelligent algorithms for the automated detection of high-risk EEG anomalies in the general population.

## 1. Introduction

Every year, over 700,000 people around the world die by suicide, a tragedy that affects families, communities, and entire countries. Suicide is a global phenomenon that involves individuals from all income groups and all regions. However, suicides can be prevented if timely, evidence-based, and personalized interventions are implemented, e.g., [1,2,3,4].

Unfortunately, the clinical assessment of the pre-suicidal state is generally based on interviews and questionnaires that most often fail to reliably identify high-risk profiles [2,3]. This is because subtle differences in meanings of words can lead to misinterpretation [5,6,7,8], and because suicidal patients often have difficulty formulating clear thoughts and emotions [9,10]. In the attempt to establish more reliable and objective methods to evaluate suicide risk, accumulating research has identified a wide range of physiological and neurological anomalies in persons with suicidal thoughts [11]. Neuroimaging studies have found that both structural and functional changes in fronto-temporal networks, prefrontal cortex, anterior cingulate cortex, and upper anterior temporal gyrus are associated with greater risk for suicide [12]. These findings support the hypothesis that the disruption of executive functions can have a direct impact on emotion regulation in individuals with suicidal behavior, impairing their ability to adaptively cope with stress [13].

On the other hand, standard neuroimaging can only detect brain anomalies at the population level and their application on a day-to-day basis still awaits validation [14]. Additionally, neuroimaging equipment, its operation, and its management are extremely costly and generally not affordable to small clinical practices and clinics [15]. In this context, several decades of research support the use of electroencephalography (EEG) as a diagnostic method in mental health [16,17]. Through electrodes appropriately placed on the scalp, EEG systems detect and record electrical signals that originate in the brain, converting them into digital form. The rhythmic EEG spectrum is typically subdivided into five main oscillation frequency bands: (1) delta (1–4 Hz), typically recorded during sleep; (2) theta (4–8 Hz), which reflects a state of drowsiness; (3) alpha (8–12 Hz), which accompanies a relaxed state; (4) beta (12–30 Hz), a common index of an engaged or active brain in healthy individuals; and (5) gamma (30–50 Hz or higher), which reflects both perception and synchronization of neural firing rates across separate brain regions [18,19]. However, while standard EEG in many cases fails to reveal biological anomalies in patients with psychiatric disorders [20], hundreds of studies support the use of quantitative EEG (qEEG), a more sophisticated method based on the application of mathematical algorithms and computer processing capabilities that allow for a comparison between the patient’s EEG activity and reference values extracted from an age and sex matched healthy population [21].

Clinical studies suggest that qEEG can be used to identify patterns of brain activity that may be associated with mental health conditions that increase the risk of suicide. In particular, reduced frontal delta power might reflect poor ability to manage psychological suffering [22], which could be in line with the evidence showing that the desire for suicide is modulated by the interaction between psychological pain and frontal delta power [23]. In addition, there is evidence indicating that depression is the most common psychiatric disorder in people who die by suicide [24]. However, while mental health professionals most often support the hypothesis that depression severity increases the risk for self-harm and suicide [25], there is research suggesting that specific electrophysiological imbalances might independently play a role in the general population [26,27]. For example, although increased theta activity is found in persons with depressive disorder [28,29,30] and has been proposed to reflect brain dysfunction in patients with depression or anxiety disorders [31,32], increased fronto-central theta power has been found to strongly correlate with behaviorally assessed SI in young healthy persons [33].

It is well known that antidepressant treatment may induce suicidal thoughts [34] and that these are associated with specific EEG changes. For example, a study investigating qEEG changes during treatment with selective serotonin reuptake inhibitors (SSRI) in MDD patients showed that left–right asymmetry of combined theta and alpha power correlated with changes in suicidal ideation from baseline [35]. Moreover, matching treatment to specific patients is too often attempted through trial and error, which may result in the worsening of their clinical profile [36,37,38] and patients with treatment resistant depression (TRD) are more likely to report hopelessness and SI [39,40], which may be associated with EEG cordance changes in midline and right frontal brain regions [41].

Although most qEEG studies on suicide have focused on the imbalances linked to suicidal thoughts, other research remarks the need to identify those factors that more specifically can lead a person to attempt suicide, questioning once again the hypothesis that suicide risk is typically associated with depression [25]. In the attempt to organize information that could be of support to the implementation of more effective suicide prevention protocols, identifying objective markers of suicide risk could be key to complement the interview-based assessments conducted by mental health professionals. In this context, suicidal ideation is more likely to occur when people’s thoughts are at rest [42]; therefore, gathering qEEG data during a resting state could provide a background on the psychological processes that take place in the mind of a person with suicidal thoughts, including negative thinking [43,44], ruminative brooding and hopelessness thoughts [44].

Our goal was to review and discuss all the resting state qEEG research conducted to date with suicidal persons, in the attempt to identify patterns of high-risk activity that mental health professionals could use to complement and potentially improve multimodal assessments in this clinical population.

## 2. Methods

A search was carried out in the Medline and Web of Science online databases for English language articles containing the terms “(eeg) AND (suicid*) AND (rest*)” without date restrictions in either the abstract, list of keywords, or both.

We looked for resting state EEG studies in children, adolescents, and adults at risk for suicide. We excluded reviews, book chapters, meeting and conference abstracts, editorial material, and publications in languages other than English. We also excluded studies in which the word “suicide” was not mentioned.

The search was concluded on 30 November 2022. It returned 40 results from Medline and 1 from Web of Science. We then applied the PRISMA method to exclude duplicates and studies not matching our inclusion criteria. The selection process yielded 7 articles (Figure 1). A descriptive summary of the papers included is presented in Table 1.

## 3. Results

In the attempt to offer a summary of high-risk resting state EEG activity, we have provided a breakdown of the imbalances found in the selected studies based on frequency bands and scalp/brain regions.

### 3.1. Delta

Greater delta power was found in patients with psychiatric disorders and with a history of SA. A medium sized study by Duan et al. [45] found that delta absolute power was greater in the frontal and central regions in psychiatric patients with a history of SA, when these were compared with either patients without a history of SA or healthy controls (HC).

### 3.2. Theta

A large qEEG study by Lee et al. [33] showed that in young healthy adults, absolute and relative theta power positively correlated with SI at all channels, except Fp1, F7, and F8. Further, the correlations of relative power at the Cz electrode site and in the fronto-central region were the most significant. Remarkably, when participants were subdivided in high and low risk groups based on their depression and anxiety scores (significantly greater in the high-risk group), the high-risk group showed higher theta relative power at F3, Fz, FCz, and Cz when compared with the low-risk group. When a hierarchical regression analysis was conducted to assess the relative strength of sex, age, and standard psychiatric questionnaires scores in predicting theta relative power in the fronto-central region, only SI scores had a significant impact, whereas measures of social connectedness, depression, or anxiety had no predictive power. However, an important limitation of the study was that participants were all young healthy adults without any psychiatric diagnoses. Therefore, these findings need confirmation in other demographics and clinical populations.

### 3.3. Alpha

A study with a medium sized cohort of female patients with depression, including patients with a history of non-suicidal self-injuries (NSSI) and patients with NSSI history combined with a history of SA (NSSI + SA), found that during eyes-closed recordings, the NSSI + SA subgroup had greater posterior alpha at P4. Additionally, EEG coherence was greater in the NSSI + SA subgroup when compared with the NSSI subgroup, especially in fronto-centro-parietal regions [46].

Similarly, Benschop et al. [47] found that within a large cohort of adult female patients with MDD, patients with SI exhibited an increase in alpha power at posterior regions when compared with non-suicidal patients.

### 3.4. Beta and Gamma

Greater absolute beta and gamma power was found in patients with mental disorders with a history of SA when compared with patients without a history of SA or HC matched for age and sex. Furthermore, phase amplitude coupling (PAC) strength between beta and gamma, as measured in fronto-central regions, was found to be weaker in persons with a history of SA when compared with persons without a history of SA or HC [45]. These findings suggest that patients with suicidal behavior exhibit weaker neuronal synchronization in the fronto-central regions of the brain.

When compared with non-suicidal depressed patients, depressed patients with a history of SA and patients with SI were found to exhibit lower frontal beta and gamma power. Moreover, greater beta and lower gamma power over the left occipital region has been found in patients with SI when compared with psychiatric controls. Finally, lower beta and gamma power over the right temporal region was found in patients with a history of SA when compared to patients with SI. In both the SI and SA groups, however, exact low resolution brain electromagnetic tomography (eLORETA) localized reduced frontal activity within the orbito-, medial-, middle-, superior-, and inferior-frontal areas and the anterior cingulate cortex. Further, the SA group exhibited reduced right temporal activity within the right inferior-, middle-, and superior-temporal cortices and the fusiform gyrus [47].

### 3.5. Frontal Alpha Asymmetry

In a large study by Rho et al. [48], frontal alpha asymmetry (FAA) was higher (increased alpha power in the left frontal region) in MDD patients with SI when compared with HC. Moreover, FAA was lower in MDD patients with SI when compared with MDD patients without SI. Interestingly, the Hamilton score for depression (HAM-D) showed a significant negative correlation with FAA in MDD patients with SI, and higher HAM-D scores were associated with relatively left skewed frontal activity. Even when clinical variables, i.e., age and sex were considered as covariates, the interaction between HAM-D scores and SI was found only in patients with MDD, implying that SI drives the effects of depressive symptom severity on FAA. Further, a significant FAA difference between low and high HAM-D scores was found only in MDD patients with SI, whereas no such difference could be observed in the group of MDD patients without SI. However, the sample size of MDD patients with SI was relatively smaller than that of either the group of HC or the group of MDD patients without SI. Additionally, the number of male participants was too small to fully determine the effect of sex.

A medium sized study by Graae et al. [27] comparing female adolescent attempters and age-/sex-matched HC found that while HC exhibited greater alpha over the right hemisphere, suicidal attempters showed a trend in the opposite direction. Furthermore, while a subsample of attempters with a diagnosis of MDD showed a trend for an inverted alpha asymmetry pattern at anterior sites, the non-depressed subgroup of attempters showed an abnormal direction of alpha asymmetry at posterior sites. Importantly, in non-depressed suicide attempters, posterior alpha asymmetry was associated with SI but not with a diagnosis of MDD.

Finally, resting state left FAA in a small cohort of patients with a history of suicidal depression was associated with an avoidance-related affective style [49]. In these patients, a decrease in relative left frontal activation and positive affective style were reported eight weeks after baseline observations.

## 4. Discussion

The papers reviewed indicate that persons at risk for suicide exhibit frequency/region specific EEG anomalies and suggest that EEG-based assessments could play an important role in predicting suicide attempts and/or death by suicide. In this context, the selected literature also suggests that the qEEG method should be regularly employed by mental health professionals to probe for brain function imbalances in patients with psychological suffering, even when depression cannot be diagnosed.

The emerging high-risk profile is generally characterized by increased slow frequency (delta and theta power) in frontal and central areas, which is in line with evidence suggesting that suicide risk is associated with reduced emotion regulation efficiency [50], and with neuroimaging research showing that altered functional abnormalities in frontal cortical areas can be associated with SA [51]. It could be proposed that increased delta activity in the brain of suicide attempters might drive a maladaptive interplay between cognitive and emotional processes, although this might be at odds with the findings showing that lower frontal delta is associated with greater psychological pain in depressed patients, which can increase the risk for suicide through hypoactivation of the default mode network (DMN) [22,23], a large-scale subset of brain regions [52,53] that is more active when the brain is at rest and less active during task performance [52]. The hypothesis of lower DMN activation coupled with increasing psychological pain might be supported by previous research suggesting a persistent state of underlying arousal that may exist in individuals with psychological pain [54]. Of note, as psychological pain is essentially an unpleasant feeling that results from consistently shifting attention towards negative thoughts, lower frontal delta power might be associated, on the one hand, with the reduced ability to reappraise or reframe pain and, on the other hand, with increased rumination about its causes and consequences [22]. Moreover, given that adaptive emotion regulation, i.e., perspective taking and reappraisal of psychological pain, might alleviate the pain and potentially restore normal DMN activity [55,56], qEEG-guided interventions aimed at normalizing delta power in frontal brain regions could play a protective role and contribute to reduce the drive for suicide in depressed persons.

The analysis of sub-frequency ranges within the alpha band may unveil the risk for SA among persons with suicidal thoughts or a history of self-harm. These sub-range anomalies can be revealed by power and coherence analyses and can extend beyond the frontal area, including the centro-parietal region [46]. In selecting these subranges, however, it is important to keep in mind how alpha activity may be influenced by the several variants of the alpha rhythm, including temporal alpha (characterized by independent alpha activity over the temporal regions, which may be found in older adults), and frontal alpha (activity over the anterior scalp regions, which may appear in relation to drugs, anesthesia, or post-sleep arousal) [19]. On the other hand, the findings indicating that psychiatric patients with greater risk for SA exhibit greater generalized absolute beta and gamma power [45] might support the hyperarousal hypothesis of psychological pain [54] and provide further evidence to propose that being psychiatrically ill may increase the risk for suicide. Conversely, the lower beta activity found in the right temporal region of persons with a history of SA when compared with persons with SI might reflect a disruption in the cognitive domain, which could be in line with previous neuroimaging research with suicide attempters indicating cortical atrophy in the temporal region of the left brain hemisphere [57]. Importantly, persons with greater risk for SA may also exhibit altered beta–gamma PAC in the fronto-central region, which might reflect aberrant emotion regulation and reduced impulse control [51], but this observation could also agree with functional connectivity evidence suggesting that the neural circuits underlying SA differ from those that underlie SI [51]. Other neuroimaging research on patients with depression and a history of SA has also shown that reduced resting state functional connectivity in the dorsolateral prefrontal cortex and DMN may correlate with increased rumination, retrieval suppression, and delay discounting, when compared with depressed patients without a history of SA [58]. Therefore, beta and gamma PAC imbalances should be accounted as high-risk factors in patients who exhibit psychological suffering related to self-referencing and impaired retrieval of painful experiences.

Finally, the literature reviewed also suggests that FAA can reflect SI in persons with MDD and/or anxiety disorders but also remarks the need to further study its role in non-depressed persons, with particular attention to sex and age-range effects. While FAA typically reflects greater alpha power in the left frontal region in depressed persons, an inverted pattern has been found in non-depressed cohorts with SI. Interestingly, anterior asymmetry may vary as a function of motivational direction rather than affective valence and greater left- than right-anterior activity may be a correlate of anger [59], which has been found to be abnormally increased in high-risk patients [60]. Hence, investigating frontal alpha imbalances between the left and right hemispheres may not only reveal an increased risk for suicide but also suggest a target for psychotherapeutic interventions. It must be remarked, however, that the frontal cortex is anatomically and functionally heterogenous and that the regions that drive FAA are yet to be fully described. In this context, employing source localization methods (e.g., eLORETA) to uncover the generators of FAA may help clarify the relationship between depression symptoms and suicidal behavior. The study by Graae et al. [27] on adolescent suicide attempters found that depressed participants exhibited an inverted pattern of the FAA that is usually found in depressed persons (greater alpha power in the left frontal region). Non-depressed attempters instead exhibited an inverted alpha asymmetry pattern at posterior regions. Moreover, the posterior alpha asymmetry in non-depressed suicide attempters correlated with SI but not with a diagnosis of depression. This EEG asymmetry distinction between depressed and non-depressed attempters may reflect separate behavioral and physiological traits, in line with the hypothesis that suicidality and depression may not share common EEG activity patterns. Interestingly, inverted FAA is typically observed in persons with aggressive behavior [61]. If SA can be viewed as aggression towards the self, the inverted asymmetry patterns seen by Graae et al. in adolescent attempters might be consistent with the view that suicidal thoughts are related not to depression but rather to aggressive behavior. However, more research in larger population samples should confirm these findings while controlling for demographic heterogeneity and factors that can further increase the risk for suicide (e.g., drug and alcohol dependence, smoking habits, increased vulnerability to stress, antidepressant therapy) [62,63,64,65,66,67].

## 5. Limitations

The present review attempted to identify selective EEG imbalances in persons with either SI or with a history of SA. However, the interpretation of results should keep into account the different methods used across studies and the heterogeneity of the populations recruited. Importantly, the evaluation of suicide risk may vary across studies as a result of the administration of different rating instruments [68] and inter-rater variability may temper the results of unstructured assessments [69]. Again, heterogeneity may also derive from the differential effects of medication and recreational drugs on both behaviorally assessed symptoms and EEG waveforms [70,71,72].

## 6. Conclusions

Resting state EEG-based evaluations of suicide risk hold promise in the implementation of more reliable suicide prevention protocols. Imbalances in frontal and left temporal brain regions might reflect abnormal activation at rest and correlate with behaviorally assessed psychological distress. Additionally, asymmetrical activation in frontal and posterior cortical regions could reflect a high-risk profile, although the pattern may be inverted in non-depressed persons. In this context, the literature reviewed supports the hypothesis proposing that SI and SA may be driven by separate neural circuits and that high-risk persons can be found within non-depressed populations, suggesting that the qEEG method could complement standard psychiatric interviews and questionnaires, potentially revealing high risk imbalances in patients displaying psychological distress but not symptoms of depression.

More research in a larger population sample should evaluate the role of the EEG anomalies outlined in the present review while also controlling for demographic heterogeneity in the attempt to identify at-risk populations and collaborate with mental health providers in implementing precision-based treatment plans, including suicide preventive strategies.

## 7. Future Directions

While a wide range of EEG methods are available to mental health professionals, the value of EEG-based assessments of suicide risk is still poorly understood. This may have a significant impact on the development and consolidation of protocols for the timely detection of high-risk brain activity. Therefore, more effort is urgently needed to facilitate the collaboration between clinicians and EEG clinics with the joint goal of developing more effective suicide prevention plans.

Given the multitude of the factors involved and the complexity of their interaction, future research should also aim at developing intelligent algorithms capable of integrating multimodal data and generating outputs that clinical staff can easily interpret and use to tailor suitable interventions. In this context, machine learning-based research combining EEG methods with the acquisition and interpretation of peripheral physiological changes holds promise [73]. Given the greater vulnerability to stress often exhibited by high-risk individuals [74] and the range of early experiences that are well known to increase suicide risk (e.g., physical, emotional, and sexual abuse) [75,76,77,78,79], resting state assessments should be complemented by ad hoc tests investigating the differential EEG changes occurring during the presentation of adequately targeted emotionally challenging stimuli. Finally, combining EEG-based methods with methods based on the investigation of sympathetic and affect changes occurring during the performance of mood induction tasks [74,80] might contribute to develop more ecological assessments and more reliably identify individuals with lower resilience.

## Figures and Tables

**Figure 1 jpm-13-00884-f001:**
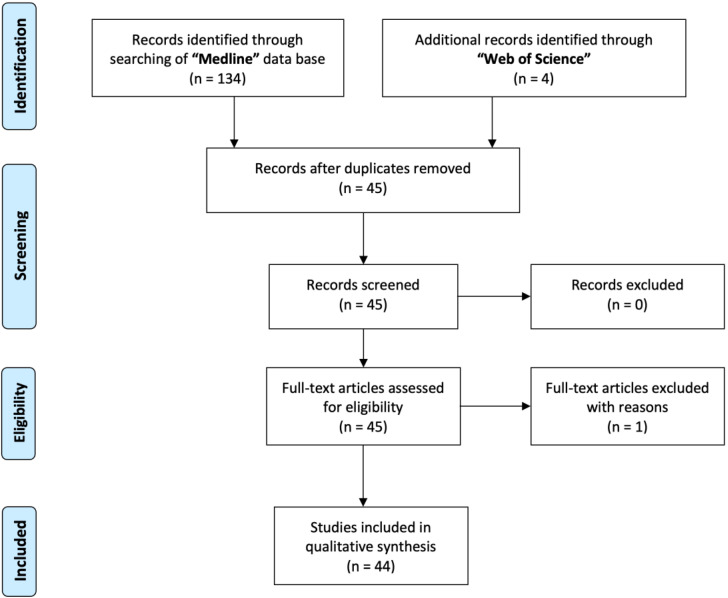
The PRISMA workflow.

**Table 1 jpm-13-00884-t001:** Summary of the reviewed studies.

Study	Population	Methods	Design	Results
[27]	Female suicide attempters (SA = 16) and matched healthy participants (HC = 22).	Eyes open and eyes closed	Mixed design. Between group cross-sectional analysis and within-subject design	1. The HC group had greater alpha power in the right hemisphere, whereas suicidal adolescents had no asymmetry in the opposite direction. 2. Alpha asymmetry over posterior regions correlated with ratings of SI but not depression severity.
[33]	Patients with higher (SSI = 33) and a lower score (SSI = 57) of suicide ideation.	Eyes closed	Mixed design. Between group cross-sectional analysis and within-subject design	1. Theta absolute and relative power in all channels was positively correlated with SSI, except Fp1, F7 and F8. 2. The high SSI group had higher theta relative power at F3, Fz, FCz, and Cz, when compared to the low SSI group. 3. Fronto-central theta relative power was greater in the high SSI group when compared to the low SSI group. 4. The SSI score was the most powerful predictor of fronto-central theta power.
[45]	Patients with a history of suicide attempt (SA = 14), no history of suicide attempt (NSA = 14), and healthy controls (HC = 14).	Eyes closed and eyes open	Between group cross-sectional analysis	1. The SA group had higher absolute delta, beta and gamma power when compared to the NSA and HC groups. 2. Fronto-central synchronization between beta and gamma power was weaker in the SA when compared to the NSA group.
[46]	Female adolescents with solely non-suicidal self-injuries (NSSI = 21), and with combined NSSI + history of suicide attempt (NSSI + SA = 24)	Eyes closed	Mixed design. Between group cross-sectional analysis and within-subject design	1. In the NSSI + SA group, the parieto–occipital power of alpha-2 (9–11 Hz) was higher and its focus was localized in the right hemisphere. 2. In the NSSI + SA group, alpha-3 (11–13 Hz) spectral power was higher than alpha-1 (8–9 Hz). 3. In the NSSI group, alpha-1 power was higher than alpha-3, 4. In the NSSI group, the foci of alpha-2 and alpha-3 were localized in the left hemisphere. 5. EEG coherence was higher in the NSSI + SA group, especially in fronto–central–parietal regions.
[47]	Female MDD patients with SA, only SI and no SI or SA (SI = 36; SA = 19; non-suicidal = 23)	Eyes open and eyes closed	Between group cross-sectional analysis	1. When compared with the non-suicidal group, the SI and SA groups had lower beta and low gamma activity in the frontal regions. 2. The SI group had increased alpha power over the posterior regions as well as increased high beta and lower gamma activity over the left occipital region when compared to the non-suicidal group. 3. The SA group had lower beta and gamma power over the right temporal region when compared to the SI group. 4. When compared with the non-suicidal group, the SI and SA groups had lower frontal activity within the orbito-, medial-, middle-, superior-, and inferior-frontal areas and the anterior cingulate cortex. 5. When compared with the non-suicidal group, the SA group had reduced right temporal activity within the right inferior-, middle-, and superior-temporal cortices and the fusiform gyrus.
[48]	Patients with major depressive disorder (MDD) with and without suicide ideation (SI), and HC (MDD = 44; MDD + SI = 23; HC = 60)	Eyes open	Mixed design. Between group cross-sectional analysis and within-subject design	1. Frontal alpha asymmetry (FAA) (reduced alpha power in the left frontal region) was lower in the MDD + SI group when compared with the MDD group. 2. Depression and anxiety symptoms were correlated with FAA only in the MDD + SI group. 3. SI moderated the effects of depressive symptom on FAA in the MDD group.
[49]	Patients with a history of suicidal depression before mindfulness-based cognitive therapy (MBCT = 10) or treatment-as-usual (TAU = 12).	Eyes open and eyes closed	Mixed design. Between group cross-sectional analysis and within-subject design	After, treatment, the TAU group showed increased prefrontal alpha asymmetry associated with a decrease in positive affective style; there was no change in the MBCT group.

## Data Availability

No original data were acquired.

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
