# Peer review of "Resting State EEG Correlates of Suicide Ideation and Suicide Attempt"

_jpm, 2023, doi:10.3390/jpm13060884_

Round 1
Reviewer 1 Report
The manuscript under review is a systematic review on the correlation between EEG resting state and suicides ideation. The study is peculiar and regards a specific and niche field. It is well-written, and the systematic research seems well-conducted. I don't think it has a great relevance in the scientific community. I also have some issues to highlight:
- The introduction is very too long (more than 3 pages!). I think the Authors should summarize/delete some paragraphs.
- In table 1, in the “Study” column, I think it should be added the surname of the fist author at least. For example, in the first line: “Duan et al. 2021 [56]”
- I think the conclusion that “that high-risk persons can be found within non-depressed populations” should be more discussed. Do the Authors think EEG could be used as a screening method? How will the used of EEG (to identify people at risk of suicide) impact the prevention strategy in the general non-depressed population? EEG is usually performed when there is the suspect of a pathology, so how the Authors think it could be used in the non-depressed population?
The are some typing errors, I guess, but the English is fine
Author Response
The manuscript under review is a systematic review on the correlation between EEG resting state and suicides ideation. The study is peculiar and regards a specific and niche field. It is well-written, and the systematic research seems well-conducted. I don't think it has a great relevance in the scientific community. I also have some issues to highlight:
- The introduction is very too long (more than 3 pages!). I think the Authors should summarize/delete some paragraphs.
We have deleted a couple of paragraphs. Now the introduction is two and a half pages. We suggest that what we left in is necessary to provide an adequate background to the study. - In table 1, in the “Study” column, I think it should be added the surname of the fist author at least. For example, in the first line: “Duan et al. 2021 [56]”
Yes, we agree with the reviewer that the suggested approach might help the reader to see the authors of the selected studies at a glance. However, references in tables usually have to conform with the referencing style required by the journal. So we're not sure that what the reviewer recommends in this very instance would be a good idea from the journal's perspective.
- I think the conclusion that “that high-risk persons can be found within non-depressed populations” should be more discussed. Do the Authors think EEG could be used as a screening method? How will the use of EEG (to identify people at risk of suicide) impact the prevention strategy in the general non-depressed population? EEG is usually performed when there is the suspect of a pathology, so how the Authors think it could be used in the non-depressed population?
As we comment in the discussion, there is evidence that psychological distress as hopelessness and despair can increase the risk for suicide. So, even when a patient cannot officially be diagnosed with depression, clinicians could use the qEEG method to investigate imbalances that might be considered as risk factors. We worded this concept as follows (in red in the conclusions):
"In this context, the literature reviewed supports the hypothesis proposing that SI and SA may be driven by separate neural circuits and that high-risk persons can be found within non-depressed populations, suggesting that the qEEG method could complement standard psychiatric interviews and questionnaires, potentially revealing hight risk imbalances in patients displaying psychological distress but not symptoms of depression."
- The are some typing errors, I guess, but the English is fine
Thank you for noticing that. We have proofread the manuscript once again and fixed the typos.
Please find the manuscript attached
Reviewer 2 Report
Dear Editor,
I really appreciate the opportunity to review the manuscript jpm-2400693 entitled:
"Resting State EEG Correlates of Suicide Ideation and Suicide Attempt"
I commend the authors for describing this critical and timely issue. The paper is interesting and well-written; however, I would like to highlight some issues that merit revision:
It is not entirely clear whether it has been investigated how, especially in young people, prior sexual abuse experienced may affect electroencephalographic alterations.
I beg the authors to emphasize this more in the manuscript, or if the data is not available to add it to the limitations.
Author Response
I really appreciate the opportunity to review the manuscript jpm-2400693 entitled: "Resting State EEG Correlates of Suicide Ideation and Suicide Attempt"
I commend the authors for describing this critical and timely issue. The paper is interesting and well-written; however, I would like to highlight some issues that merit revision:
It is not entirely clear whether it has been investigated how, especially in young people, prior sexual abuse experienced may affect electroencephalographic alterations.
I beg the authors to emphasize this more in the manuscript, or if the data is not available to add it to the limitations.
Thank you for remarking how sexual abuse can increase the risk for suicide in persons of all age ranges. However, none of all the papers selected for review was specifically conducted on such population.
Ad advised, we have included a few lines (and references) in the “Future directions” section to mention how much still needs to be done to establish high risk EEG markers of suicide in populations with a history of emotional, physical or sexual abuse”.
Round 2
Reviewer 1 Report
I think the introduction is still a little bit too long. The paper is adressed to experts and I think some general information could be deleted. However, I agree to publish the paper as it is after the revisions the authors have already done.